# EMT cells increase breast cancer metastasis via paracrine GLI activation in neighbouring tumour cells

Deepika Neelakantan[1,2], Hengbo Zhou[1,3], Michael U.J. Oliphant[1,4], Xiaomei Zhang[5], Lukas M. Simon[6], David M. Henke[7], Chad A. Shaw[7], Meng-Fen Wu[5], Susan G. Hilsenbeck[5,8], Lisa D. White[7,9], Michael T. Lewis[5,9] & Heide L. Ford[1,2,3,4]

Recent fate-mapping studies concluded that EMT is not required for metastasis of carcinomas. Here we challenge this conclusion by showing that these studies failed to account for possible crosstalk between EMT and non-EMT cells that promotes dissemination of non-EMT cells. In breast cancer models, EMT cells induce increased metastasis of weakly metastatic, non-EMT tumour cells in a paracrine manner, in part by non-cell autonomous activation of the GLI transcription factor. Treatment with GANT61, a GLI1/2 inhibitor, but not with IPI 926, a Smoothened inhibitor, blocks this effect and inhibits growth in PDX models. In human breast tumours, the EMT-transcription factors strongly correlate with activated Hedgehog/GLI signalling but not with the Hh ligands. Our findings indicate that EMT contributes to metastasis via non-cell autonomous effects that activate the Hh pathway. Although all Hh inhibitors may act against tumours with canonical Hh/GLI signalling, only GLI inhibitors would act against non-canonical EMT-induced GLI activation.

[1] Department of Pharmacology, University of Colorado–Denver, 12800 East 19th Avenue, Room P18-6115, Aurora, Colorado 80045, USA. [2] Molecular Biology Program, University of Colorado Anschutz Medical Campus, Aurora, Colorado 80045, USA. [3] Cancer Biology Program, University of Colorado Anschutz Medical Campus, Aurora, Colorado 80045, USA. [4] Integrated Physiology Program, University of Colorado Anschutz Medical Campus, Aurora, Colorado 80045, USA. [5] Lester and Sue Smith Breast Center, Baylor College of Medicine, Houston, Texas 77030, USA. [6] Institute of Computational Biology, Helmholtz Zentrum München (GmbH), Neuherberg 85764, Germany. [7] Department of Molecular and Human Genetics, Baylor College of Medicine, Houston, Texas 77030, USA. [8] Department of Medicine, Baylor College of Medicine, Houston, Texas 77030, USA. [9] Departments of Molecular and Cellular Biology and Radiology, Baylor College of Medicine, Houston, Texas 77030, USA. Correspondence and requests for materials should be addressed to M.T.L. (email: mtlewis@bcm.edu) or to H.L.F. (email: heide.ford@ucdenver.edu).

In recent years, immunohistochemical analyses and multiplex high-throughput single cell sequencing of human tumour cells have shown that tumours are composed of diverse cell subpopulations containing different driver mutations, gene and protein expression profiles, growth rates and responses to chemotherapeutics[1,2]. Such heterogeneity is exacerbated by cellular plasticity, where some cells may undergo oncogenic epithelial-to-mesenchymal transition (EMT), resulting in loss of cell–cell adhesion and polarity, as well as reduced epithelial and elevated mesenchymal protein expression[3,4], increased migration and invasion, and enhanced dissemination from the primary tumour[3]. As metastases in patients appear epithelial[3], the reverse process, mesenchymal-to-epithelial transition, may occur to allow tumour cell colonization in secondary metastatic sites[5], establishing cellular plasticity as an important aspect of tumour progression.

However, the role of EMT in carcinoma metastasis is controversial. Recent lineage-tracing studies argue against the requirement of EMT for metastasis, as reporter-tagged cells that underwent a previous EMT were not found at the secondary site[6,7]. However, these studies did not address the potential cooperation between EMT and non-EMT cells during the metastatic process, as EMT cancer cells may enable non-EMT cells to gain access to the secondary site, leading to macrometastatic growth[1]. Thus, metastasis can be influenced by intratumoural heterogeneity: where a small proportion of primary tumour cells that have undergone an EMT[4,6] may influence neighbouring, non-EMT tumour cells.

Twist1, Snail1 and Six1 are EMT-inducing transcription factors (EMT-TFs) that have all been associated with breast cancer metastasis[4,8]. All three EMT-TFs regulate critical developmental processes such as cell survival, migration and invasion, in part by influencing EMT[4,9]. In addition, they are downregulated post embryogenesis, but re-expressed in various cancers where they cell autonomously induce EMT, resulting in increased percentages of tumour-initiating cells and enhanced metastasis[10,11]. In carcinomas, Twist1 and Snail1 transcriptionally repress E-Cadherin (E-Cad) and upregulate mesenchymal genes[4]. Similarly, Six1 overexpression induces EMT by regulating E-Cad localization and altering other EMT markers[10].

During development and cancer, EMT-TFs act in concert with several signalling networks including transforming growth factor-β, Wnt and Hedgehog (Hh)[1,4]. The Hh signalling pathway is a prominent regulator of embryonic development, where Hh ligands function as morphogens to control numerous developmental processes[12]. Interestingly, in *Drosophila* eye development, *hh* is a direct target of *sine oculis* (the homologue of Six1)[13] and Six1 regulates Hh/GLI signalling during lung development and in fibroblasts[14,15]. In addition, Twist1 and Hh/GLI signalling are intimately linked during development[16], and recently Twist1 and Snail1 were associated with the Hh pathway in tumour-initiating cells[17,18].

In mammals, canonical activation of Hh/GLI signalling involves binding of one the Hh ligands, Desert Hh (DHH), Indian Hh (IHH) or Sonic Hh (SHH) to Patched-1 (PTCH1) or PTCH2 receptors, relieving the inhibitory activity of PTCH on Smoothened (SMO). When inhibition is relieved, levels of the transcriptional activator forms of one or more GLI TFs (GLI1, 2 or 3) increase in the nucleus, resulting in activation of Hh target genes[12]. Non-canonical activation of the GLI TFs can occur in a Hh- or SMO-independent manner via secreted factors such as transforming growth factor-β[19]. Importantly, autocrine and paracrine Hh-mediated cross-talk between tumour cells and the tumour microenvironment[20] results in increased proliferation, stem cell self-renewal and metastasis in various cancers[21]. In basal cell carcinoma (BCC) and medulloblastoma, activated Hh

signalling is often due to mutations in pathway components such as *PTCH* and *SMO*, whereas in other tumour types including breast, mutations are not observed at high frequency. Instead there is evidence for Hh ligand-dependent pathway hyperactivity[15,22,23]. As numerous studies link Hh signalling to progression in multiple tumour types, derivatives of cyclopamine (for example, GDC-0449 and IPI926), a naturally occurring plant-derived steroidal alkaloid which targets SMO, are in clinical trials for select patients with BCC and medulloblastoma, and are proving to be efficacious[24,25]. However, these inhibitors have not shown promise in breast cancer[26] despite evidence for activation of this pathway[23,27].

Herein, we demonstrate that EMT-TFs Twist1, Snail1 and Six1 influence neighbouring carcinoma cells in a non-cell autonomous (NCA) manner, by increasing EMT features and aggressive properties of cells not expressing these TFs. Six1 is a key mediator of the NCA effects downstream of Twist1 and Snail1, and can induce metastasis of non-Six1 expressing, non-EMT cells. All three EMT-TFs function non-cell autonomously by activation of GLI-mediated transcription in non-EMT cells, but employ different mechanisms of pathway activation, some of which are Hh ligand and SMO independent. We find that pharmacological inhibiton of GLI, but not SMO, in the non-EMT cells efficiently inhibits the NCA phenotypes imparted by all three EMT-TFs. Importantly, we demonstrate that in selected patient-derived breast cancer xenograft (PDX) models, GANT61 (a GLI1/2 inhibitor) inhibits tumour growth, whereas IPI926 (a SMO inhibitor) does not. Taken together, our data suggest that upstream SMO inhibitors may not prove efficacious in tumours where a proportion of cells activate GLI via EMT-TFs and instead argue that inhibitors directly targeting GLI may be more effective.

## Results

**EMT-TFs impart metastatic properties on neighbouring cells.** To mimic the primary tumour, where only a small percentage of cells may express EMT-TFs[4], we co-cultured GFP+ HMLER-Control (HMLER-Ctrl) cells with tRFP+ HMLER-Snail1 or Twist1 cells in a 1:1 or 10:1 ratio and performed migration assays. Co-culture with either HMLER-Snail1 or Twist1 cells increased HMLER-Ctrl cell migration, even when only one of ten cells in the mixture expressed the EMT-TF (Fig. 1a–d). Studies show Twist1 and Snail1 can take around 14 days to cell autonomously induce EMT[11,28]. To determine whether EMT can be induced non-cell autonomously in this system, GFP+ HMLER-Ctrl cells were co-cultured with tRFP+ Twist1/Snail1 cells for 14–16 days at a 1:1 and 10:1 ratio, cells were sorted using flow cytometry and protein was extracted to measure EMT markers. Surprisingly, no morphologic changes were observed nor was a decrease in E-cad or cytokeratin 18 (epithelial markers), or increase in vimentin (mesenchymal marker) observed, when HMLER-Ctrl cells were co-cultured with EMT-TF-expressing cells (Supplementary Fig. 1a,b). Thus, Twist1 and Snail1 non-cell autonomously increase aggressive properties in neighbouring cells without causing EMT in those cells, suggesting that EMT and metastatic properties are not always linked downstream of these EMT-TFs.

To test whether the NCA effects of Twist1 and Snail1 were due to the activity of a secreted factor, conditioned medium (CM) was isolated from HMLER-Snail1/Twist1 and placed on HMLER-Ctrl cells. Indeed, HMLER-Ctrl cells had increased migration (Fig. 1e–h) in the presence of CM from HMLER-Snail1/Twist1-expressing cells as well as invasion (Supplementary Fig. 1c–f). The invasion assays were performed for 16–18 h, as no proliferation differences were observed in HMLER-Ctrl cells receiving different CM over this time period (Supplementary Fig. 1g,h).

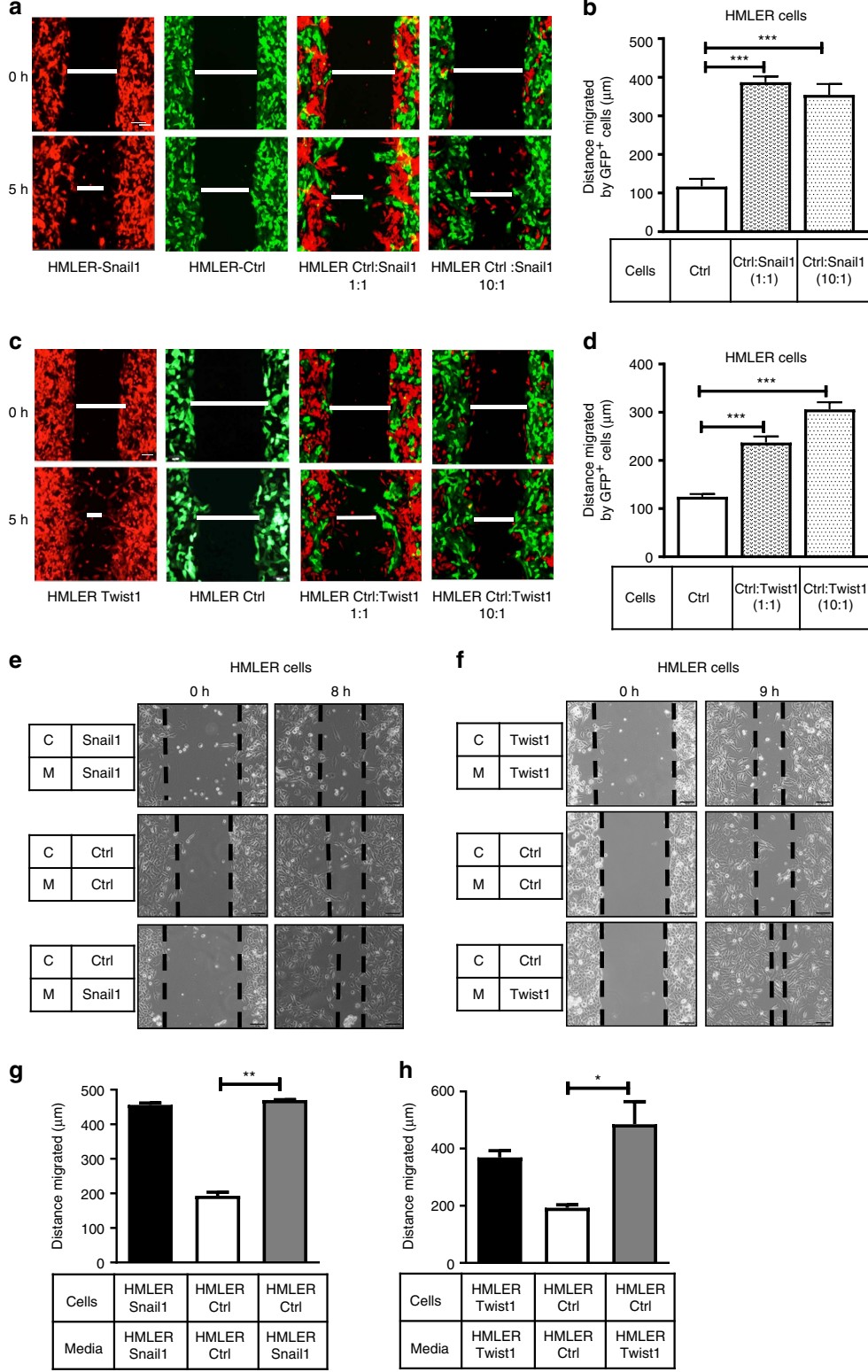

**Figure 1 | Snail1 and Twist1 non-cell autonomously increase metastatic properties of control cells.** Five hour migration assay performed on (**a,c**) tRFP[+] HMLER-Snail1/Twist1 cells alone, GFP[+] HMLER-Ctrl cells alone, or a mixture of the two cell types at 1:1 or 10:1 ratio. (**b,d**) Quantification of GFP[+] cell migration in **a,c**. (**e,f**) Representative 8–9 h migration assays of HMLER-Ctrl cells cultured in CM from (**e**) HMLER-Snail1 cells and (**f**) HMLER-Twist1 cells; C-cells, M-Media. (**g,h**) Quantification of cell migration in **e,f**. Scale bar, 100 μm, s.e.m. shown, $n \geq 3$. One-way ANOVA with Tukey's post test in all cases. $*P < 0.05$, $**P < 0.01$ and $***P < 0.001$.

**Six1 acts downstream of EMT-TFs to mediate NCA phenotypes.** Studies show that EMT-inducing molecules and TFs often affect the levels of each other in tumour cells[4,29]. In line with this observation, we found that Six1 messenger RNA (Supplementary Fig. 2a) and protein expression (Fig. 2a) is increased in HMLER-Snail1 and HMLER-Twist1 cells compared to HMLER-Ctrl cells. Therefore, to

examine whether Six1 is required downstream of Twist1 and/or Snail1 to mediate their NCA effects, we performed Six1 knockdown (KD) in HMLER-Snail1 and Twist1 cells using small interfering RNAs (consistently achieving 70–80% Six1 KD on the mRNA and protein levels (Supplementary Fig. 2a and Fig. 2a)). We then cultured HMLER-Ctrl cells in CM from HMLER-Snail1/Twist1 cells ± Six1 KD and performed migration and invasion assays. Six1 KD in HMLER-Snail1 and Twist1 cells dramatically inhibited the NCA stimulation of migration (Fig. 2b–e) and invasion (Supplementary Fig. 2b–e) in HMLER-Ctrl cells. Notably, Six1 levels in HMLER-Ctrl cells receiving CM (from cells ± Six1KD) remained low and unchanged when CM was transferred to cells (Supplementary Fig. 2f), demonstrating that the observed effects were due to Six1 KD in HMLER-Twist1/Snail1 cells from which the CM was derived. Thus, Six1 is necessary downstream of Twist1 and Snail1 to non-cell autonomously increase 'metastatic' properties of non-TF-expressing cells.

**Six1 non-cell autonomously induces EMT-like phenotypes.** To test whether Six1 is sufficient to mediate NCA phenotypes in vitro, we used the MCF7 model where Six1 stable overexpression (Supplementary Fig. 2g) has been shown to cell autonomously induce EMT and metastasis[8]. It should be noted that Six1 expression in this model does not alter expression of Twist1 or Snail1, and thus it can be studied in a context separate from Twist1 and Snail1 (Supplementary Fig. 2h,i). As MCF7-Six1 cells do not have increased in vitro migration and invasion when compared to MCF7-Ctrl cells, other 'EMT parameters' known to be affected by Six1 cell autonomously were examined to determine whether Six1 also has NCA functions. In contrast to HMLER cells, MCF7-Control (Ctrl) cells cultured in MCF7-Six1 CM underwent a shift in their EMT protein expression profile, where cytokeratin-18 (CK18), an epithelial marker, was decreased, and Fibronectin (FN1), a mesenchymal marker, was increased (Fig. 2f). Furthermore, membranous E-Cad was strikingly downregulated in MCF7-Ctrl cells cultured in Six1 CM (Fig. 2g) compared to when cultured in Ctrl CM. Finally, we found that MCF7-Six1 cells have increased anoikis resistance compared with MCF7-Ctrl cells, and that MCF7-Ctrl cells cultured in Six1 CM gained this resistance phenotype (Fig. 2h). Interestingly, NCA phenotypes are transferrable across cell types, as HMLER-Ctrl cells cultured in CM from MCF7-Six1 cells showed increased migration as compared with when cultured in CM from MCF7-Ctrl cells (Supplementary Fig. 2j), suggesting that the CM contains soluble factors that influence the receiving cells, although the receiving cells may in part dictate the response. Nevertheless, although particular molecular events may vary in different backgrounds, Six1 is both necessary and sufficient to impart aggressive characteristics on non-EMT-TF-expressing cells.

**Hh ligands are differentially regulated by EMT-TFs.** The Hh/GLI signalling pathway is known to function with all three EMT-TFs during development and cancer[17,30], and as previously discussed, hh is a direct target of sine oculis during development[13]. Thus, we tested whether Hh ligands, SHH, DHH or IHH were upregulated in HMLER and MCF7 cells. HMLER-Ctrl and HMLER-Twist1/Snail1 ± siSix1 cells did not express differential mRNA levels of any Hh ligand (Fig. 3a and Supplementary Fig. 3a,b) and secreted SHH in CM from HMLER-Ctrl and HMLER-Snail1/Twist1 ± Six1KD cells was below the detectable threshold (Fig. 3b). In contrast, increased SHH mRNA and protein, and increased secreted SHH was observed in MCF7-Six1 versus MCF7-Ctrl cells and in MCF7-Six1 CM (Fig. 3c–e). There were no changes in IHH or DHH levels between MCF7-Ctrl and

Six1 cells (Supplementary Fig. 3c,d). Although no increase in Hh ligands was observed in HMLER-Twist1/Snail1 cells, the effects of Six1 on SHH levels are not limited to MCF7 cells. Indeed, endogenous Six1 regulates SHH expression in A2780 ovarian cancer cells (Supplementary Fig. 3e). Interestingly, Six1 overexpression in HMLER-Ctrl cells (in the absence of Twist1 or Snail1) causes SHH induction only in the presence of its critical co-factor, Eya2 (Supplementary Fig. 3f)[31]. As the HMLER cells express lower baseline levels of Eya2 compared with MCF7 cells (Supplementary Fig. 3g), these data suggest that the ability of Six1 to induce Hh ligands may be dependent on sufficient availability of an Eya cofactor in the system.

**EMT-TFs activate GLI via various mechanisms.** As Hh/GLI signalling can also be activated in a Hh ligand-independent, non-canonical, manner[32] and as activation of the pathway has been associated with all three EMT-TFs[15,17,18], we further examined whether GLI activity was increased in cells cultured in Snail1/Twist1 CM, despite the absence of Hh ligands. To this end, a GLI1-specific lentiviral reporter containing seven GLI1 consensus-binding sites fused to GFP (7-Gli1) or a mutant GLI (m-Gli1)-GFP reporter was transfected into HMLER-Ctrl cells. Surprisingly, HMLER-Ctrl cells cultured in Snail1/Twist1 CM displayed a significant increase in 7-Gli1-GFP, but not m-Gli1-GFP, reporter activity compared to when cultured in Ctrl CM and this effect was abrogated in cells cultured in HMLER-Snail1/Twist1 + Six1KD CM (Fig. 4a,b and Supplementary Fig. 4a,b). Furthermore, HMLER-Ctrl cells cultured in HMLER-Snail1/Twist1 CM had increased expression of numerous Hh pathway target genes, which was dependent on Six1 expression (Fig. 4c,d and Supplementary Fig. 4c). Similarly, MCF7-Ctrl cells cultured in MCF7-Six1 CM exhibited significantly increased 7-Gli1 (but not m-Gli1) reporter activity, as well as increased expression of Hh pathway genes, compared to when cultured in Ctrl CM (Fig. 4e,f and Supplementary Fig. 4d). As expected, recombinant SHH added to the medium on MCF7-Ctrl cells significantly activated the pathway, demonstrating specific activation of the reporter in response to Hh pathway activation (Supplementary Fig. 4e). Thus, Six1 is sufficient and necessary downstream of Snail1 and Twist1 to non-cell autonomously activate Hh/GLI signalling in non-EMT-TF-expressing cells. Furthermore, NCA GLI activation by EMT regulators can occur via both Hh ligand-dependent and -independent mechanisms, an effect that is likely to be due to cellular context and/or the particular combination of EMT regulators.

The finding that GLI is non-cell autonomously activated by EMT-TFs via different mechanisms suggests that the type of inhibitor used to target the Hh pathway will affect response differentially. Thus, we inhibited the Hh pathway using 5E1, a function-blocking monoclonal antibody targeting Hh ligands[33], cyclopamine (an upstream Hh pathway inhibitor targeting SMO) or GANT61 (a downstream Hh pathway inhibitor targeting GLI1/2 (ref. 34)). As we did not detect Hh ligands in the HMLER system, only cyclopamine and GANT61 were used to differentiate between the involvement of SMO versus activation bypassing SMO. Only GANT61, and not cyclopamine, inhibited 7-GLI activation when HMLER-Ctrl cells were cultured in Snail1/Twist1 CM (Fig. 4g and Supplementary Fig. 4f). In contrast, all three inhibitors significantly, and equally, decreased NCA activation of 7-GLI, but not m-GLI, activation in MCF7-Ctrl cells cultured in MCF7-Six1 CM (Fig. 4h and Supplementary Fig. 4g). Together these data suggest that Hh signalling via GLI can be non-cell autonomously activated by Six1, Twist1 and Snail1 in all settings, but that the mechanism of pathway activation differs in each context.

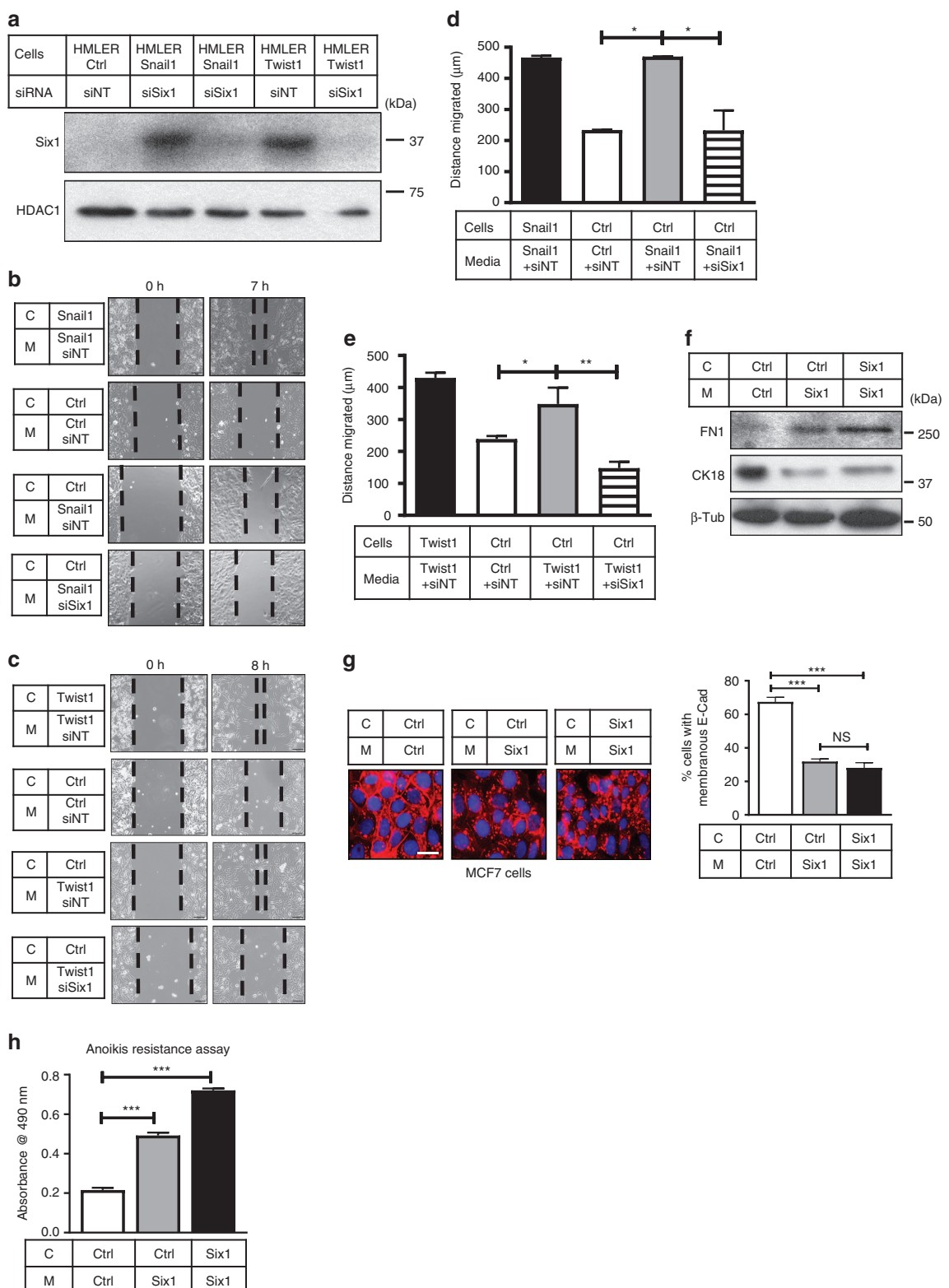

**Figure 2 | Six1 is necessary (downstream of Snail1/Twist1) and sufficient to mediate NCA effects.** (**a**) Western blot analyses performed on WCLs from HMLER-Ctrl, HMLER-Snail1 and HMLER-Twist1 cells transfected with 150 nM of siSix1 or a non-targeting small interfering RNA (siRNA) pool (siNT). HDAC1, loading control. (**b,c**) Representative 7–8 h migration assay of HMLER-Ctrl cells in CM from HMLER-Snail1 or Twist1 cells ± siSix1. (**d,e**) Quantification of cell migration in **b,c**. (**f**) Western blot analyses performed on WCLs from MCF7-Ctrl and MCF7-Six1 cells cultured in indicated CM for 48 h. β-Tub, β-Tubulin, loading control; CK18, cytokeratin 18; FN1, fibronectin. (**g**) Representative ICC of E-Cad (red) in MCF7-Ctrl and Six1 cells cultured in indicated CM for 48 h (DAPI, blue). Quantification of % membranous E-Cad, n ≥ 100; scale bar, 20 μm. (**h**) Representative anoikis resistance graph (plotted as absorbance as a measure of cell number) of MCF7-Ctrl or MCF7-Six1 cells cultured in indicated CM for 24 h. C-cells, M-Media; Scale bar, 100 μm, s.e.m. shown, n ≥ 3, one-way ANOVA with Tukey's post test in all cases; NS, not significant; *P < 0.05, **P < 0.01 and ***P < 0.001.

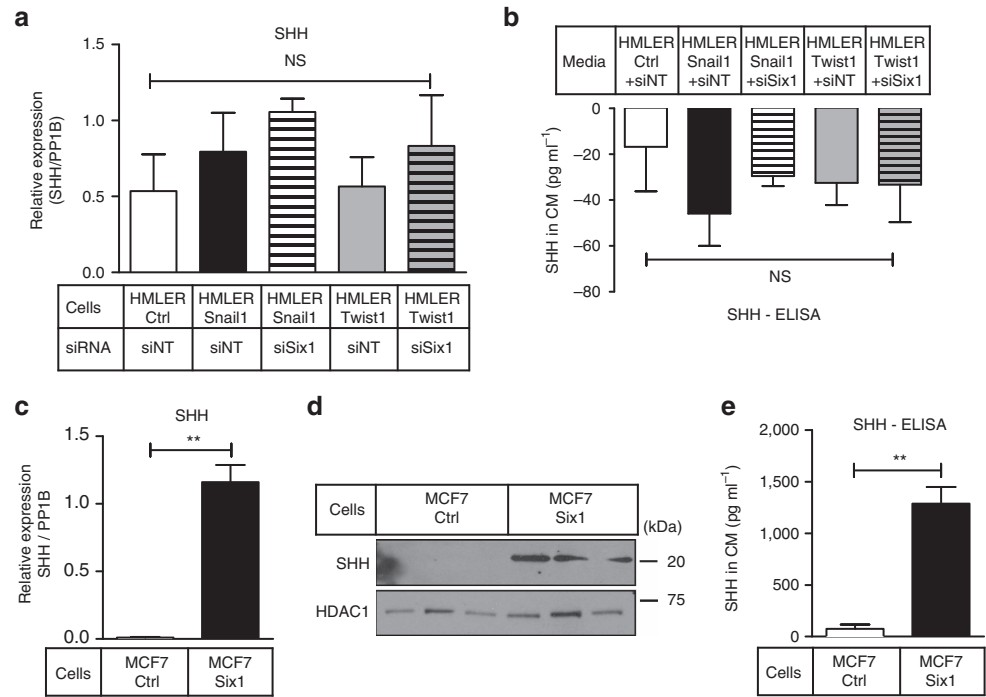

**Figure 3 | SHH is upregulated downstream of Six1 in specific contexts.** (**a**) Quantitative real-time PCR (qRT–PCR) analysis (on mRNA isolated from cells) and (**b**) ELISA analysis on CM from HMLER-Ctrl, Snail1 and Twist1 cells transfected with 150 nM of siSix1 or siNT. (**c**) qRT–PCR analysis on mRNA isolated from three different MCF7-Ctrl versus MCF7-Six1 clones (clones of same cell type combined) normalized to PP1B, (**d**) western blot analysis on WCLs using antibodies against SHH and HDAC1 as a loading control, (**e**) ELISA on CM from three different clones of MCF7-Ctrl and MCF7-Six1 cells (clones combined in **e**). s.e.m. shown, compiled experiments $n \geq 3$. One-way ANOVA with Tukey's post test for **a,b** and two-tailed unpaired $t$-test for **c,e**. NS, not significant; \*\*$P$-value$< 0.01$.

**GLI is the key mediator for the NCA function of EMT-TFs.** To determine whether targeting GLI in recipient cells would abrogate the NCA phenotypes mediated by EMT-TFs, migration and invasion assays were performed with HMLER cells in the absence or presence of cyclopamine or GANT61. Cyclopamine treatment of HMLER-Ctrl cells had no influence on the ability of HMLER-Snail1/Twist1 CM to increase migration or invasion (Fig. 5a and Supplementary Fig. 5a). In contrast, GANT61 treatment of HMLER-Ctrl cells cultured in HMLER-Snail1/ Twist1 CM abolished NCA increases in migration and invasion (Fig. 5b and Supplementary Fig. 5b). Treatment of MCF7-Ctrl cells with either cyclopamine or 5E1 caused a reversal of alterations in EMT markers that were observed when MCF7-Ctrl cells were incubated in MCF7-Six1 CM (Fig. 5c). Membranous E-Cad was also robustly restored with the use of these inhibitors (Fig. 5d,e and Supplementary Fig. 5c,d), although anoikis resistance was not (Supplementary Fig. 5e). Importantly, MCF7-Ctrl cells cultured in Six1 CM and treated with GANT61 showed a restoration in the CK18 and FN1 protein levels back to control levels (Fig. 5f), robust rescue of E-Cad on the cell membranes (Fig. 5g,h) and, interestingly, a rescue in anoikis sensitivity (Fig. 5i). These data demonstrate that inhibiting Hh signalling using GANT61 is an effective means to inhibit NCA effects of EMT-TFs in all contexts.

**Inhibition of GLI suppresses NCA-mediated metastasis.** To determine whether Six1, the central EMT-TF involved in the described NCA phenotypes, also increases metastasis non-cell autonomously *in vivo*, we tagged MCF7-Ctrl and MCF7-Six1 cells with a luciferase and tRFP vector, respectively, (MCF7-Ctrl-luc and MCF7-Six1-tRFP) and orthotopically either 'singly' injected or co-injected them into immunocompromised mice. 'Singly injected' MCF7-Ctrl cells contained a 1:1 mixture of MCF7-Ctrl-

luc and untagged cells, and 'singly injected' MCF7-Six1 cells contained a 1:1 mixture of MCF7-Six1-tRFP and untagged cells. This experimental strategy carefully controls for the tagged cell numbers, when comparing with the 1:1 mixture of MCF7-Ctrl-luc and MCF7-Six1-tRFP cells, a condition referred to as 'mixed tumours' (Fig. 6a). In mice bearing similar tumour volumes, we found a significant increase in distant luminescence signal of MCF7-Ctrl-luc (non-EMT) cells when they were co-injected with MCF7-Six1 (EMT) cells compared to when 'singly' injected (Fig. 6b,c), indicating that Six1 is able to non-cell autonomously increase the metastasis of non-Six1 cells *in vivo*. To determine whether Hh/GLI signalling and, specifically, activation of GLI was necessary downstream of Six1 for increased NCA metastasis, we inhibited GLI activity using GANT61. Mice that had mixed tumours were randomized once their tumour volumes reached 1 cm³, and half were treated every other day for 18 days with 50 mg kg⁻¹ of GANT61 and the other half with vehicle control (Fig. 6a). The increased distant MCF7-Ctrl-luc signal observed in the vehicle treated mixed tumour groups was dramatically decreased upon GANT61 treatment (Fig. 6b,d,e). We also observed a decrease in luciferase signal (from non-EMT cells) in the primary tumours of mice treated with GANT61 compared with vehicle (Fig. 6b,f,g). Surprisingly, EMT cells also grew and metastasized more efficiently in the presence of non-EMT cells (Supplementary Fig. 6a,b). Interestingly, GANT61 treatment modestly (but significantly) inhibited the metastatic ability of the tRFP-labelled EMT cells themselves (Supplementary Fig. 6a,c,d), but did not significantly affect their growth in the primary tumour (Supplementary Fig. 6a,e,f), nor did GANT61 treatment significantly affect overall primary tumour volume when measured using calipers (Supplementary Fig. 6g). These data suggest that EMT cells do not depend on GLI to the same degree as non-EMT cells, but that GLI signalling is important in both

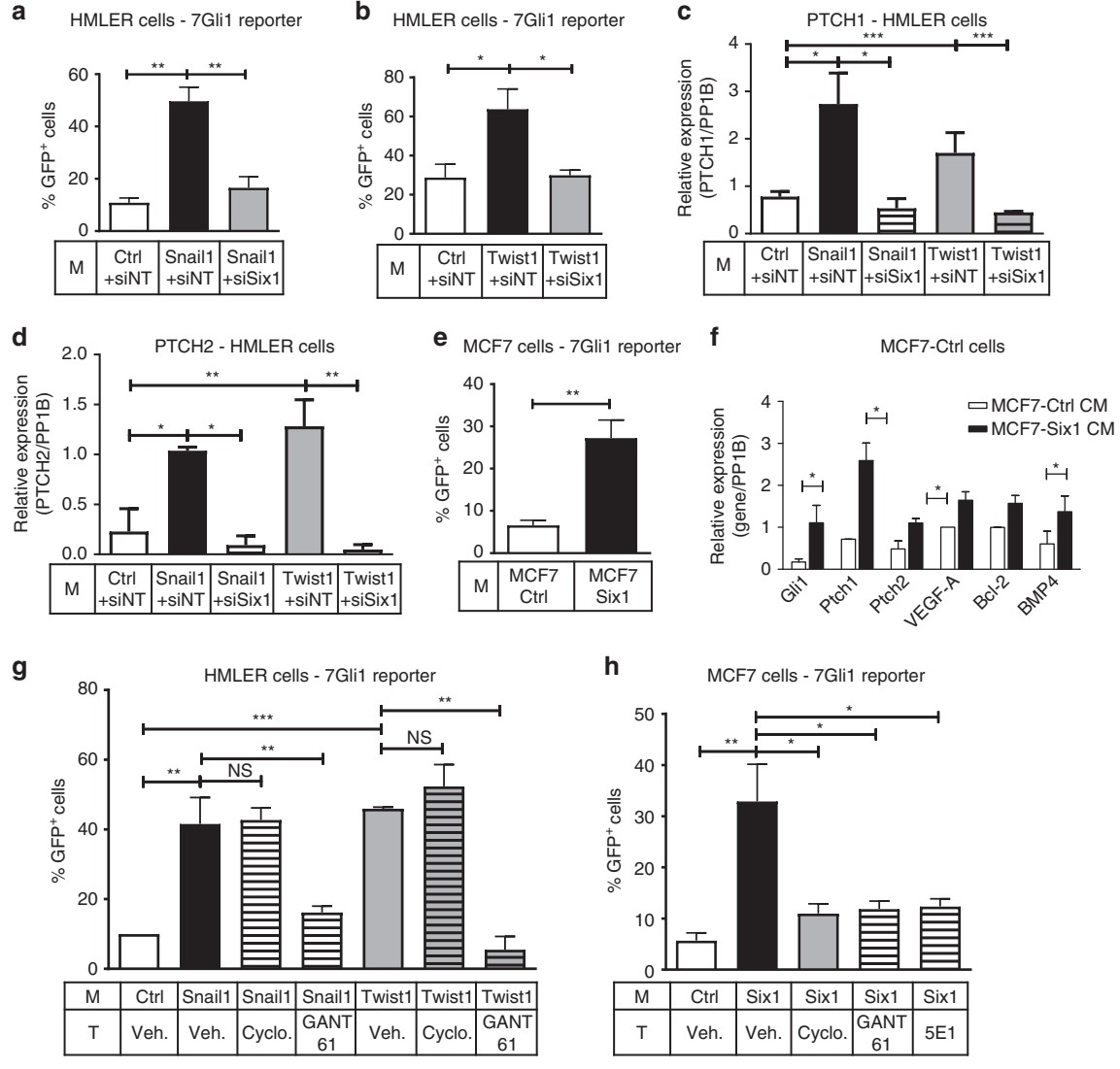

**Figure 4 | Hh/GLI signalling is activated canonically and non-canonically downstream of EMT-TFs.** 7-Gli1 reporter assays performed in (**a,b**) HMLER-Ctrl cells treated with CM from HMLER-Ctrl cells or from HMLER-Snail1/Twist1 cells ± siSix1. (**c,d**) Quantitative real-time PCR (qRT–PCR) analyses of Hh pathway target genes in HMLER-Ctrl cells cultured in indicated CM. (**e**) 7-Gli1 reporter assays in MCF7-Ctrl cells cultured in MCF7-Ctrl or Six1 CM. (**f**) qRT–PCR analyses of Hh pathway target genes in MCF7-Ctrl cells cultured in indicated CM. (**g,h**) 7-Gli1 reporter assays of (**g**) HMLER cells cultured in CM from HMLER-Ctrl, Snail1 and Twist1 cells, and (**h**) MCF7-Ctrl cells cultured in MCF7-Ctrl/Six1 CM and treated with vehicle (Veh.), 10 μM cyclopamine, 10 μM GANT61, or 3–5 μg ml$^{-1}$ of 5E1 monoclonal antibody. Gene expression is normalized to PP1B; all Gli1-GFP assays represented as %GFP$^+$ cells. M, Media; T, Treatment; s.e.m. shown, $n \geq 3$, except for **c,d,f** using different sets of CM where $n \geq 2$. One-way ANOVA with Tukey's post test for (**a–d,g,h**) and two tailed unpaired $t$-test for (**e,f**). NS, not significant; $*P < 0.05$, $**P < 0.01$ and $***P < 0.001$.

cell types to mediate metastasis. Most importantly, the data demonstrate that EMT and non-EMT cells can cooperate to increase metastasis in heterogeneous tumours, and that inhibition of Hh signalling using GANT61 can inhibit both the growth and NCA metastasis of non-EMT cells.

**GANT61 decreases tumour growth in PDX models with EMT.** We next asked whether the relationship of EMT-TFs and GLI was relevant to human breast cancers. We found strong positive correlations between *SIX1*, *SNAI1* or *TWIST1* with *GLI1* (a Hh pathway target) in numerous breast cancer data sets. However, a consistent positive correlation between EMT-TFs and Hh ligands was not observed in those same data sets, again suggesting non-canonical mechanisms of activation of the Hh/GLI pathway by EMT-TFs (Fig. 7a). Furthermore, increased levels of each of the EMT-TFs, together with high *GLI1*,

also correlated more significantly with worsened prognosis (relapse-free survival and distant metastasis-free survival) in breast cancer patients spanning numerous grades and subtypes, when compared with the EMT-TFs or *GLI1* alone (Fig. 7b–d and Supplementary Table 1).

Next, we performed RNA sequencing analysis on previously established breast PDX models[35] and interrogated expression data to determine whether PDX models similarly showed a correlation between expression of EMT-TFs and GLI activation. PDX were first ranked by composite expression of *TWIST1*, *SNAI1* and *SIX1*, and then evaluated for expression of each *GLI* gene and associated Hh network genes. Consistent with previous results, increased GLI activity was observed in PDX tumours expressing EMT-TFs and, surprisingly, did not correlate with Hh ligand levels (Fig. 8a). Interestingly, of the EMT-TFs, Six1, the central mediator downstream of Twist1 and Snail1, significantly correlated with percent of PDX-bearing mice

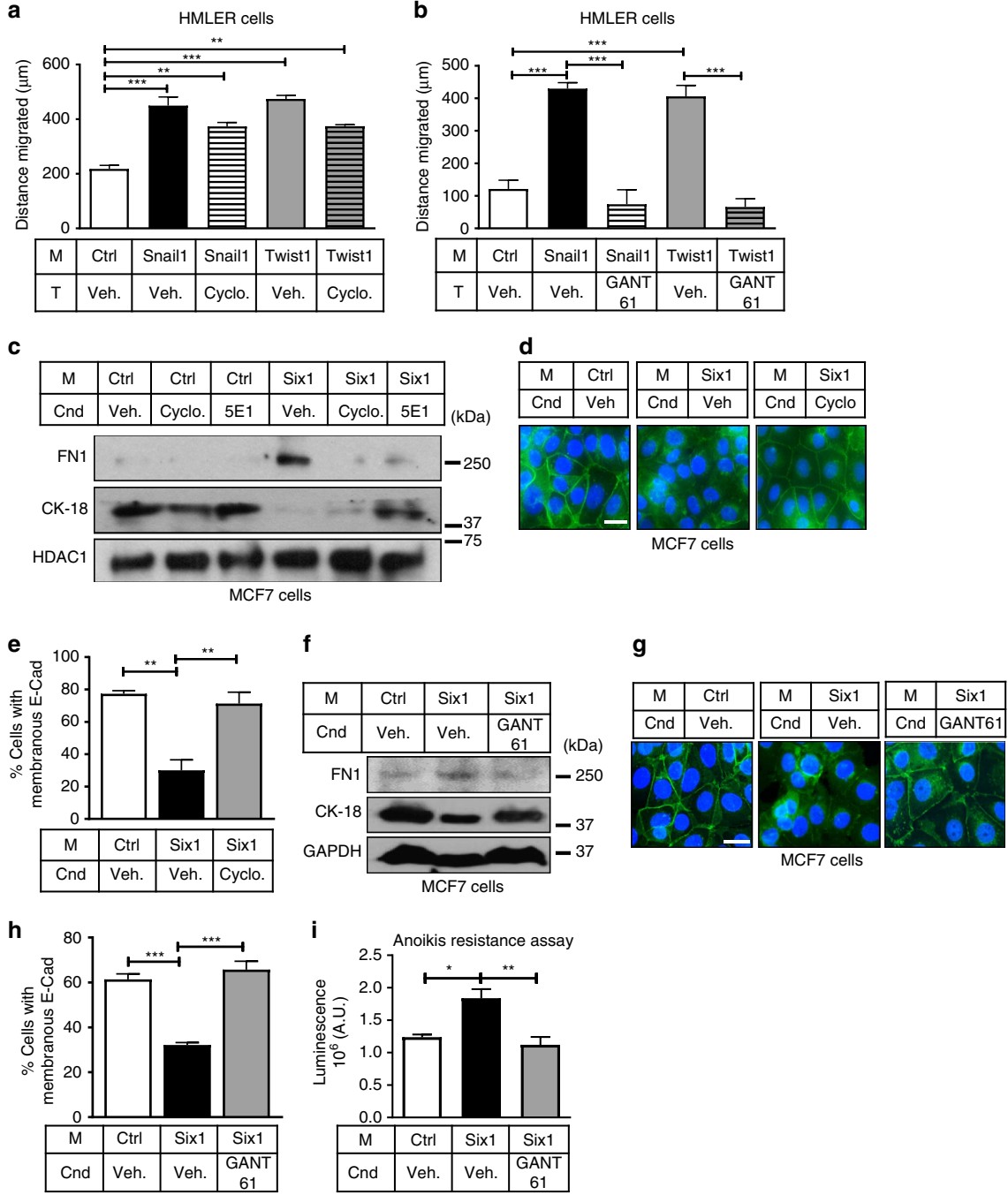

**Figure 5 | NCA activation of GLI is central downstream of EMT-TFs to alter phenotypes.** (**a**,**b**) Seven hour migration assays of HMLER-Ctrl cells cultured in HMLER-Snail1/Twist1 CM treated with (**a**) cyclopamine (Cyclo.) or (**b**) GANT61 or corresponding vehicle (Veh.) control. (**c**,**f**) Western blot analyses of WCLs from MCF7-Ctrl cells cultured in MCF7-Ctrl/Six1 CM for 48 h ± indicated inhibitors. HDAC1, GAPDH, loading controls. (**d**,**g**) Representative ICC of E-Cad (green) in MCF7-Ctrl cells cultured in MCF7-Ctrl/Six1 CM for 48 h ± indicated inhibitors. Scale bar, 20 µm; DAPI, blue (**e**,**h**) Quantification of % membranous E-Cad in **d**,**g**. Counts performed on ≥ 45–60 cells per condition. (**i**) Anoikis resistance assay in MCF7-Ctrl cells in indicated CM ± GANT61 for 24 h. Cnd, Condition; M, Media; T, Treatment; s.e.m. shown, $n \geq 3$, one-way ANOVA with Tukey's post test in all cases. *$P < 0.05$, **$P < 0.01$ and ***$P < 0.001$.

in which circulating tumour cells were found, suggesting a potential key role for this EMT-TF in metastatic dissemination (Fig. 8b).

To test the hypothesis that breast tumours with activated GLI signalling are more susceptible to GLI inhibitors compared with upstream inhibitors targeting SMO, we chose two models, MC1 and BCM-2147. MC1 was used previously to support a role for Hh signalling in tumour-initiating cell regulation and shows

elevated *GLI2* expression. BCM-2147 shows high expression of EMT-TFs, as well as *PTCH1* and *PTCH2*, suggestive of activated Hh signalling, but does not show high Hh ligand expression. In tumour-bearing mice treated with either IPI926 or GANT61 for 14 days (versus vehicle), only GANT61 caused significant growth inhibition (Fig. 8c). These data are consistent with the observation that EMT-TFs converge on Hh/GLI pathway activation independent of Hh ligands and suggest that targeting

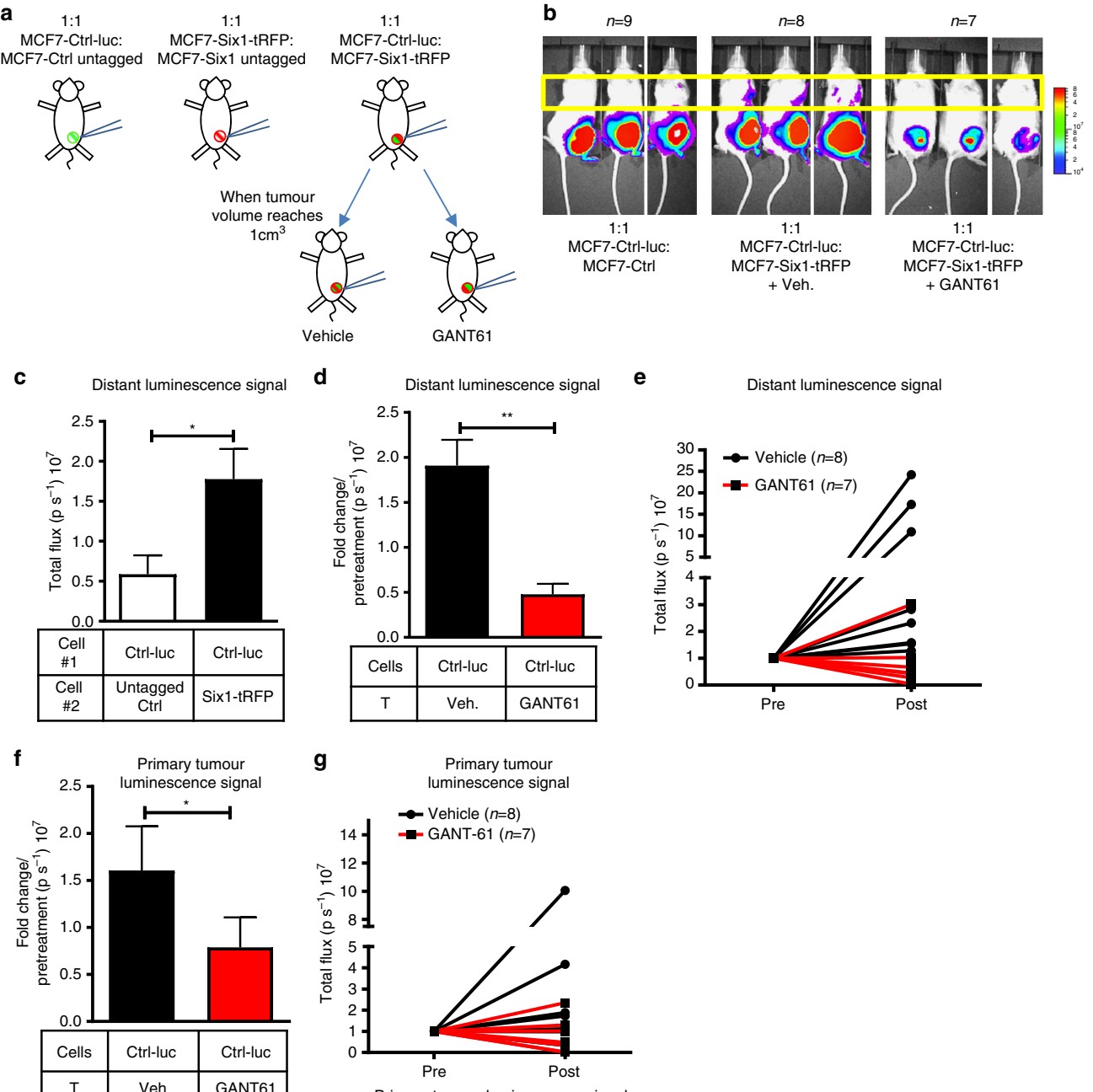

**Figure 6 | GANT61 inhibits NCA increase in metastasis *in vivo*.** (**a**) Diagrammatic representation of the metastasis experiment. Female NOG/SCID mice were orthotopically injected in the fourth mammary fat pad with either MCF7-Ctrl-luc/MCF7-Ctrl-untagged (1:1) or MCF7-Six1-tRFP/MCF7-Six1-untagged cells (1:1) or co-injected with MCF7-Ctrl-luc and MCF7-Six1-tRFP cells (1:1). Mixed tumours were treated with 50 mg kg$^{-1}$ of GANT61 or vehicle (Veh.) every other day for 18 days once tumour volume reached 1 cm$^3$. (**b**) Representative bioluminescent imaging of mice (detecting ONLY the MCF7-Ctrl-luc cells) performed at similar tumour volumes for MCF7-Ctrl-luc and mixed injection groups ± GANT61 treatment. (**c**) Quantification of specifically the MCF7-Ctrl-luc signal (in lymph nodes/lungs, yellow boxed region in **b**) in MCF7-Ctrl-luc and mixed injection + vehicle treatment groups; represented as p s$^{-1}$, photons per second. (**d,f**) Quantification of specifically the MCF7-Ctrl-luc signal from (**d**) distant sites and (**f**) primary tumour in mixed tumours groups +/− GANT61 treatment, represented as fold change over pre-treatment signal. (**e,g**) Normalized luminescent signal from (**e**) distant sites and (**g**) primary tumours of individual mice pre- and post treatment. T, Treatment; s.e.m. shown, two-tailed unpaired *t*-test for all cases, *$P < 0.05$ and **$P < 0.01$.

Hh signalling through downstream effectors such as GLI may be more efficacious in breast tumours expressing EMT-TFs.

## Discussion
In a heterogeneous primary tumour made up of various subpopulations of cells, data show that only a small percentage of epithelial tumour cells undergo an EMT and/or have metastatic potential at any one time[1,4,6]. Although undergoing EMT is thought to allow cells to better metastasize[36], recent studies argue that oncogenic EMT is not a requirement for metastasis[6,7]. However, these studies do not address the possible interactions that may occur between EMT and non-EMT cells in a heterogeneous primary tumour, resulting in metastasis of either cell type.

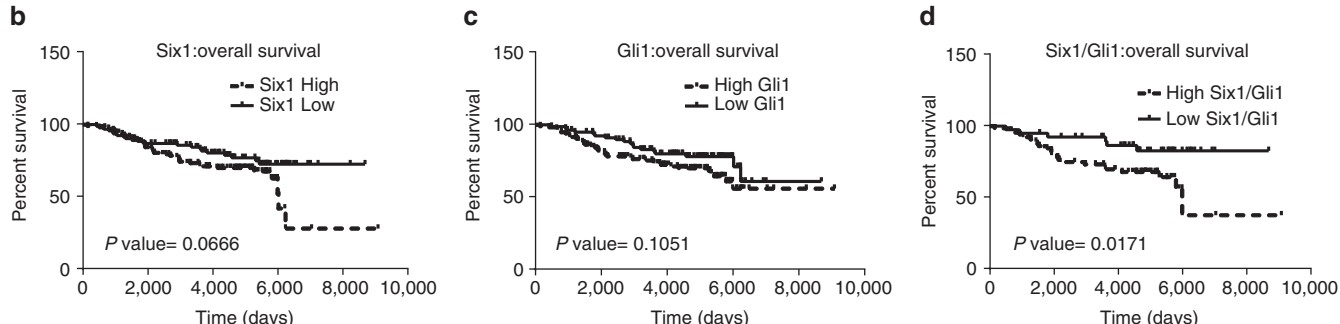

**a**

| EMT-TF | Data set name (breast) | Positive correlations (with P value) | | | |
| --- | --- | --- | --- | --- | --- |
| | | GLI | IHH | DHH | SHH |
| Six1 | Gluck | < 0.0001 | NS | NS | NS |
| | Stickeler | 0.0010 | No probe | NS | NS |
| | Tabchy | < 0.0001 | NS | No probe | 0.0066 |
| | Curtis | < 0.0001 | NS | NS | NS |
| | Shankavaram cell line 2 | 0.0473 | NS | NS | NS |
| Snai1 | Esserman | 0.0154 | No probe | NS | NS |
| | TCGA | 0.0373 | NS | NS | NS |
| | Hatzis | < 0.0001 | < 0.0001 | No probe | < 0.0001 |
| | Miller 2 | 0.0259 | NS | No probe | NS |
| Twist1 | TCGA | < 0.0001 | NS | NS | NS |
| | Bittner | < 0.0001 | NS | NS | NS |
| | Stickeler | 0.0056 | No probe | NS | NS |
| | Gluck | < 0.0001 | 0.0085 | NS | NS |

**Figure 7 | Expression of GLI1 and EMT-TFs positively correlate in human patients.** (**a**) Table demonstrating positive correlation between GLI1 mRNA and *Six1*, *Snai1* and *Twist1*, but not between EMT-TF mRNA and Hh ligand mRNA, in multiple breast cancer datasets. Data obtained from Oncomine. (**b–d**) High levels of both GLI1 and Six1 together significantly correlate with worsened prognosis; data obtained from GSE7390.

We demonstrated that prominent EMT-TFs, Snail1 and Twist1, non-cell autonomously increase aggressive properties of non-TF-expressing cells. Six1 is upregulated and is required downstream of Snail1 and Twist1 to mediate their NCA phenotypes, and is itself sufficient to non-cell autonomously increase aggressive properties of cells lacking Six1. Interestingly, EMT cells can enhance metastatic properties in cells without causing them to undergo EMT, as the NCA effects in different systems did not always alter EMT markers, but in all cases, enhanced properties associated with EMT and increased aggressiveness. Hence, fate mapping of EMT cells alone may not truly reflect their role in metastatic dissemination, as in all cases examined, the aggressiveness of epithelial cells (non-EMT; measured by varying parameters) is increased by the presence of and/or CM from EMT cells.

Importantly, all three EMT-TFs converge onto Hh/GLI signalling to non-cell autonomously increase aggressive properties of non-TF-expressing cells. However, their mode of Hh pathway activation differs, despite the fact that Six1 is a key driver of NCA activation of GLI in all contexts. Hh signalling is activated in cancers canonically (via Hh ligands) and non-canonically via GLI (independent of SMO)[32]. Our data show that in some contexts, Six1 non-cell autonomously activates GLI by upregulating and secreting high levels of SHH. Hence, in MCF7 cells where Six1 is the main EMT mediator, most of its NCA effects are abrogated efficiently using both upstream inhibitors and downstream inhibitors of the pathway. Interestingly, in HMLER cells, where Twist1 and Snail1 are the main drivers of EMT (although dependent on Six1), SHH levels were unaltered and NCA phenotypes were only abrogated with GANT61, indicating SMO-independent non-canonical activation of GLI in this context.

Several non-canonical mechanisms of GLI1/2 activation have been described in the literature including, but not limited to, mechanisms that promote transcription of GLI1/2, increase their stability or regulate their cellular compartmentalization[32]. In addition, various molecules have been linked to GLI activation[32] in a non-canonical manner, which can be explored as potential means by which the EMT-TFs non-canonically activate Hh signalling. To date, we have not identified a known non-canonical pathway by which EMT-TFs non-cell autonomously activate Hh signalling, and thus are currently using unbiased approaches to answer this question. Taken together, our data suggest that the context of EMT-TF

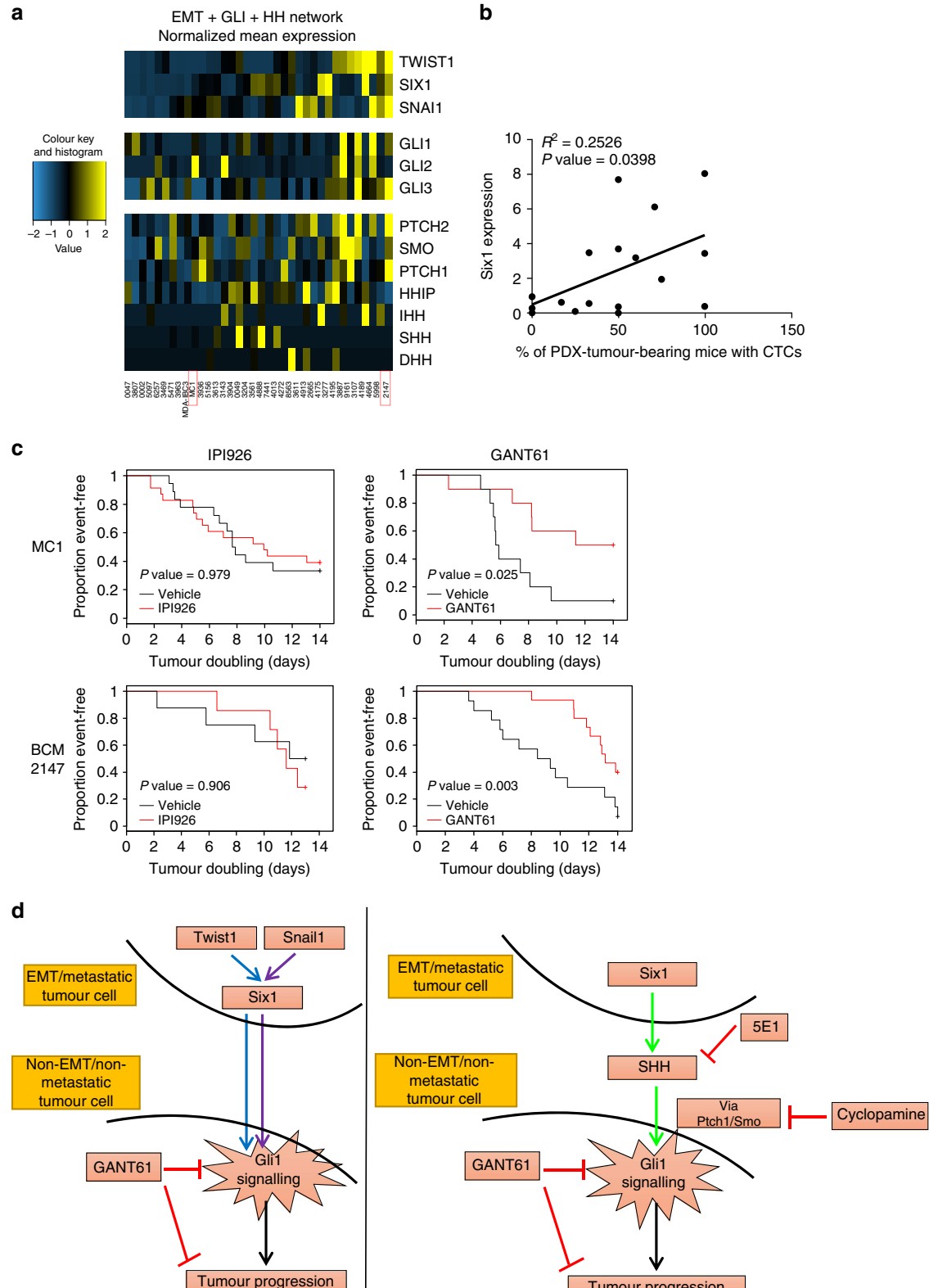

**Figure 8 | GANT61 but not IPI926 suppresses tumour growth in PDX models.** (**a**) Heat-map generated from RNA-seq analyses performed on PDX tumours (red boxes-PDX models treated with drugs). (**b**) Linear regression analysis performed on PDX tumours shows a positive correlation between Six1 mRNA expression and the percentage of PDX-tumour-bearing mice with circulating tumour cells (CTCs). (**c**) Kaplan–Meier curves depicting tumour doubling times for MC1 and BCM-2147 PDX models treated with IPI926 and GANT61. Comparisons between IPI926/GANT61 and vehicles were made using Wilcoxon test. (**d**) Graphical summary of paper which demonstrates that EMT-TFs can induce GLI activation in neighbouring cells not expressing the EMT-TFs, and that this activation can occur either via canonical or non-canonical means. In either case, the activation of GLI1 in neighbouring cells results in the attainment of aggressive properties, leading to tumour progression which can be inhibited by GANT61.

expression and perhaps particular combination of EMT-TFs, dictates how GLI is activated. Nonetheless, EMT-TFs uniformly converge on GLI activation (Fig. 8d) and thus our data have implications for targeting heterogeneous tumours in which EMT occurs in subset of cells.

Importantly, we found that GANT61, which targets GLI1/2 directly, and thus acts downstream of SMO, effectively decreased the metastasis of MCF7-Ctrl cells induced by MCF7-Six1 cells *in vivo*. GANT61 also had a selective effect on the growth of MCF7-Ctrl cells in the mixed tumours, which could be an additional means by which decreased metastasis of MCF7-Ctrl cells is seen in the mixed tumours with GANT61 treatment. However, the effect of GANT61 on MCF7-Ctrl cells in the primary mixed tumours is less than that on the MCF7-Ctrl cells in the metastases of mixed tumours, suggesting that GANT61 may work via both inhibiting growth as well as active metastasis of these cells. GANT61 also had a modest, but significant, effect on MCF7-Six1 cell metastasis. As the bulk of an epithelial carcinoma is expected to be non-EMT cells, drugs such as GANT61 may be efficacious in limiting both primary tumour growth and metastasis. However, our data suggest that residual EMT cells may survive such treatment in the primary tumour and thus a combination of GANT61 with additional therapies may be more efficient in targeting both EMT and non-EMT cells in primary heterogeneous tumours. As Hh signalling is also active in the tumour microenvironment[20], effects of GANT61 are likely to be due to influences on both tumour and stromal cells.

Each EMT-TF positively correlates with *GLI1*, across multiple breast cancer data sets, but not consistently with Hh ligands. Interestingly, in PDX models, activated Hh/GLI signalling correlates more robustly with EMT-TF expression than with Hh ligand expression, suggesting that EMT-TFs in breast cancer can non-canonically activate Hh/GLI signalling independent of Hh ligand. These results are in contrast to the current paradigm in breast cancer where Hh signalling is prominently thought to be activated in a Hh ligand-dependent manner. Moreover, PDX tumours were more sensitive to growth inhibition by GANT61 versus IPI926 under this treatment regimen, suggesting that targeting GLI may be more efficacious than targeting SMO. However, as the drugs are given systemically, it is possible that they are also affecting tumour growth via influences on the microenvironment or on other tissues.

Derivatives of cyclopamine that target SMO, such as vismodegib (GDC-0449), as well as other SMO inhibitors such as sonidegib (LDE-225), are currently in clinical trials for patients with BCC and medulloblastoma[24,25]. Although SMO inhibitors show promising results in these patients, they have been less efficacious as single agents in other solid tumours such as breast cancer, despite evidence for activated Hh signalling[12]. Our data provide a possible explanation for why SMO inhibitors have not been successful in advanced stage breast cancer patients[26], as these patients, particularly if their tumours express EMT-TFs, may activate Hh signalling in a small number of cells, and/or in a SMO-independent manner. Instead, downstream GLI-targeting inhibitors may be more powerful in this context. Studies also show that SMO antagonist treatment often leads to drug resistance in patients via mutations that render SMO incapable of being bound by the drugs, or by Hh pathway activation via GLI in a SMO-independent manner[37,38]. This finding, combined with our data which demonstrate that GLI signalling is activated downstream of EMT-TFs both canonically and non-canonically, makes a strong case for use of downstream antagonists like GANT61. GANT61 antagonists would be expected to more effectively inhibit Hh signalling in the context of heterogeneous breast cancers and thus inhibit both primary tumour growth and metastasis.

## Methods

**Cell culture and plasmids.** HMLER derivative cell lines were a generous gift from Dr Robert Weinberg (Massachusetts Institute of Technology, 2009). MCF7-Ctrl and MCF7-Six1 cells were generated as previously described[8] and cultured according to ATCC recommendations. A2780 cells were a kind gift from Dr Jennifer Richer (University of Colorado Anschutz Medical Campus). Three clones each of the MCF7-Ctrl and MCF7-Six1 cells were used in some experiments as indicated in figure legends. Gli1 reporter assays were performed with all three clones which showed similar GLI activation and as they expressed similar levels of Six1 and SHH, one of each clone was used in other assays. Cell lines were routinely checked for mycoplasma and, if found positive, were either treated or earlier mycoplasma-negative freeze downs were used for experiments. The lines were profiled via short tandem repeat profiling to confirm their identity (February 2011, April 2015). Transient KD of Six1 was performed using ON-TARGETplus SMARTpool small interfering RNAs (L-020093-00-0020, Dharmacon). Cyclopamine (C-8700, LC Labs), GANT61 (G9048, Sigma) and 5E1 (Developmental Studies Hybridoma Bank, The University of Iowa) were used at 5 or 10 µM and 3–5 µg ml$^{-1}$, respectively, in each experiment. GANT61 used in the mixed tumour experiment was purchased from MedChem Express (HY-13901) and resuspended in 4:1 corn oil: 100% Ethanol mixture. For the PDX models, GANT61 was synthesized by Dr Rune Toftgard (Karolinska Institutet) and IPI926 was a kind gift from Infinity Pharmaceuticals (Cambridge, MA). rhSHH (1845-SH, R&D systems) was used at 1 µg ml$^{-1}$ in experiments. Vehicle treatment in experiments—combination or single treatment of 100% Ethanol (for cyclopamine), dimethylsulfoxide (for GANT61) and NS-1 (control supernatant for 5E1) depending on experiment. Cells were tagged with pLenti NS-tRFP or pLNXC2-Zsgreen. MCF7-Ctrl-luc cells were generated using SFG-nes-TGL-luciferase plasmid as described previously[8]. pcDNA3.1 hygro vectors were used for transient overexpression of Six1 and Eya2. For conditioned medium (CM) collection, equal numbers of cells were plated and allowed to grow for 24 h. The next day, the cells were washed and replaced with fresh medium (serum-free or serum-containing medium). CM was collected from the cells 48 h later, filtered through 0.45 µm filter and stored at −20 °C. Repeated freeze-thaws of CM was avoided.

**Gli1-GFP reporter assays.** Cells ($3–5 \times 10^4$) plated in 24-well plates in different CM and/or drug conditions were simultaneously transfected in different wells with either 7-Gli1-GFP or m-Gli1-GFP[39] reporters. Cells in each condition were also transfected with a constitutively expressing GFP-Ctrl vector on the same vector background in a separate well. Forty-eight hours post transfection, the number of GFP$^+$ cells in each condition (indicating activated GLI signalling) was counted using a fluorescence microscope and normalized to the number of GFP$^+$ cells in the well containing GFP-Ctrl vector to account for differences in transfection efficiencies of cells cultured in different CM and/or drugs. Results from three or more independent experiments using different sets of CM were grouped and graphed as % of 7-Gli1 or m-Gli1 GFP$^+$ cells. Further details on the reporters will be described elsewhere.

**Quantitative real-time PCR.** Total RNA was extracted using the RNAeasy RNA isolation kit (Qiagen). Complementary DNA synthesis was performed using iScript (Biorad) from 1 µg of mRNA. Quantitative real-time PCR assays were performed using ssoFast Evagreen supermix (BioRad), and run and analysed using the Biorad CFX96. The primer sequences used are listed in Supplementary Table 2.

**Immunoblot analysis.** Whole-cell lysates (WCL) and nuclear extracts were generated as previously described[8]. In brief, RIPA buffer was used to extract WCLs and NE-PER Nuclear and Cytoplasmic Extraction Reagents (78833, ThermoFisher) were used for nuclear extracts. In both cases, equal amounts of lysates (35–50 µg) were electrophoresed and transferred to polyvinylidene difluoride (PVDF) membranes. The membranes were blocked in 5% milk in TBST for 1 h and incubated with primary antibody at 4 °C O/N. The antibodies used and dilutions are listed in Supplementary Table 3. For co-culture experiments for EMT markers, HMLER-Ctrl GFP$^+$ cells were cultured with HMLER-Snail1 or Twist1 tRFP$^+$ cells in 10:1 or 1:1 ratio and individually for 14–16 days. The GFP$^+$ cells (and tRFP$^+$ cells in the control conditions) were then obtained from each condition by flow cytometry and WCLs were extracted from them to use in western blot analysis. Uncropped western blottings are shown in Supplementary Fig. 7.

**Immunocytochemistry.** MCF7 cells (10 k) were plated in each well of eight-well chamber slides (154534, Nunc Lab-Tek) with the different CM/drug conditions and incubated for 48 h. The cells were fixed using 4% paraformaldehyde and stained with primary antibodies at 4 °C O/N. Slides were mounted using ProLong Gold Antifade Mountant with 4,6-diamidino-2-phenylindole (P-36931, ThermoFisher) and images were taken using a fluorescence microscope from field of vision containing cells at about 60–80% confluency, as it was observed that E-cadherin expression in MCF7 cells was substantially affected by cell confluence. Blinded membranous E-Cad quantification was performed by dividing the number of cells with membranous E-Cad by the total number of 4,6-diamidino-2-phenylindole (DAPI)-stained cells in that field of vision to obtain

a percentage ($n > 100$). The antibody and dilution used are listed in Supplementary Table 3

**Anoikis resistance assay.** Cells were cultured in the different CM for 48 h, after which they were trypsinized, counted and 50,000 cells were plated on poly-HEMA (12 mg ml$^{-1}$ in 95% Ethanol) coated plates for 24 h in CM. The next day, cells were retrieved and re-plated in 96-well plates in full media for 5–6 h until they attached. The surviving cells were then analysed using either MTS (3-(4,5-dimethylthiazol-2-yl)-5-(3-carboxymethoxyphenyl)-2-(4-sulfophenyl)-2H-tetrazolium and phenazine methosulfate) assay (Promega) or crystal violet staining to determine their anoikis resistance.

**Cell migration/wound-healing assay.** Cell migration was measured using a modified scratch assay with culture inserts that create a uniform 500 μm gap (80209, Ibidi). Cells ($6 \times 10^5$ cells per ml) were plated in different CM in each compartment of the insert and incubated O/N. Cells were treated with the drugs or vehicle at the time of CM addition. Inserts were removed after 16–18 h and distance migrated by cells in 5–8 h was measured using DP2-BSW software (v2.2; Olympus).

**Cell invasion assay.** Cell culture inserts (353097, BD Falcon) were coated with 1:20 diluted Matrigel in serum-free media. Fifty thousand cells were plated on the matrigel in different CM and/or drug conditions (added when CM was added to cells) and allowed to invade through the membrane towards medium containing 10% FBS in the bottom chamber. After 16–18 h, cells and Matrigel above the membrane were wiped with cotton swabs, and cells below the membrane were fixed in 4% PFA and stained with crystal violet. The numbers of invading cells were analysed either by measuring their absorbance or by counting the cells using a bright-field microscope.

**Cell proliferation assay.** HMLER cells were plated to equal confluence in 96-well plates in five replicates (in different CM) for 24 h. Day 0 time point was analysed 4–5 h post plating of cells and day 1 time point, at 24 h post plating. The cells were analysed using either MTS or CellTiter-Glo assays (Promega). The values obtained for day 1 were normalized to the respective day 0 values (set at 1). For MTS assay, the cells were incubated with MTS reagents added to each well as per the manufacturer's instructions. Plates were incubated at 37 °C for 2–4 h and absorbance was measured 490 nm. For CellTiter-Glo assays, cells were incubated with kit reagents at room temperature for 30 min and luminescence was measured using a luminometer.

**Enzyme linked immunosorbent assay.** Cells were cultured in serum-free medium and CM was collected after 48 h. In the case of MCF7 cells, CM was collected from three clonal isolates of each cell type (MCF7-Ctrl and Six1). Equal volumes of samples were loaded on the SHH ELISA plate (ab100639, Abcam). The concentration of SHH in the CM was determined using the generated standard curve, after normalizing the absorbance of the sample loaded to the concentration of total protein in the sample, which was previously measured using Lowry assay.

Each *in-vitro* experiment was independently and successfully repeated more than three times in the laboratory.

**Mouse models.** All animal studies were performed according to protocols reviewed and approved by the Institutional Animal Care and Use Committee at the University of Colorado AMC and Baylor College of Medicine. Power calculation analysis based on pilot *in vivo* experiments were performed before main animal experiments to determine mouse numbers. The mice were randomized into treatment groups once the tumours reached a certain size, depending on the study. The studies were not conducted in a blinded manner.

In mixed tumour experiments, oestrogen pellets were implanted subcutaneously in mice a day before injection with tumour cells as described previously[40].

**Luciferase/tRFP experiment.** MCF7 cells were injected orthotopically in 100 μl of a 1:1 mixture of sterile PBS: growth-factor reduced Matrigel (354230, Corning) in the nipple of the fourth mammary fat pad in 32 NOG/SCID female mice <1 year of age. The conditions were as follows: 1:1 ratio of 500,000 MCF7-Ctrl-luc:500,000 untagged MCF7-Ctrl cells, 1:1 ratio of 500,000 MCF7-Six1-tRFP:500,000 untagged MCF7-Six1 cells and 1:1 ratio of 500,000 MCF7-Ctrl-luc:500,000 MCF7-Six1-tRFP cells. Tumour volume (length $\times$ (width$^2$) $\times$ 0.4) and metastatic progression were followed weekly by measuring luminescence and fluorescence signal using IVIS imaging software. Mice receiving mixed injections were randomly divided into two groups and treated either with GANT61 or vehicle (4:1, corn oil: 100% ethanol), every other day for 18 days with 50 mg kg$^{-1}$ of GANT61 (or vehicle) sub-cutaneously in the supra-scapular region, once tumours reached 1 cm$^3$. Metastatic spread of MCF7-Ctrl-luc cells in the singly injected and mixed tumours was compared and analysed between mouse groups with similar tumour volumes. Metastatic spread of MCF7-Six1-tRFP cells was compared and analysed at the same time point, corresponding to similar tumour volumes across different injection

groups. Luminescence and fluorescent signal from all sites across mice groups was compared pre- and post treatment, and analysed using Living Image software. Singly injected and mixed tumour mice were killed when tumour volume reached 2 cm$^3$ and at the end of treatment, respectively.

**PDX models.** PDX tumour fragments were transplanted into the cleared fat pad of 3- to 4-week-old female SCID/Beige mice. Mice were treated either with vehicle, IPI926 (40 mg kg$^{-1}$, oral gavage) or GANT61 (50 mg kg$^{-1}$, subcutaneous injection) once a day for 2 weeks, starting from a tumour volume of $\approx$200 mm$^3$. Tumour volume (mm$^3$) was measured twice weekly and calculated as length $\times$ (width$^2$) $\times$ 0.5. Circulating tumour cells in the PDX models were collected using Rarecyte and a combination of manufacturer's protocols and those used previously[41].

**RNA-seq and analysis.** Total RNA was extracted from PDX samples and 10 ng was used to generate and amplify whole transcriptome cDNA (NuGen Ovation v2, NuGen Technologies). Three grams of cDNA from each sample was fragmented to 250–400 bases using the Covaris S2 focused ultrasonicator (Covaris). Using the Illumina TruSeq DNA-Seq library preparation kit (Illumina Technologies), a double-stranded DNA library was generated with 1 g of the sheared cDNA. The library was quantified with the Kapa quantitative PCR Library Quantitation Kit (Kapa). DNA library (11 pM) was loaded onto an Illumina HiSeq 2000 FlowCell and clusters generated on the Illumina cBot. The libraries were sequenced on the HiSeq 2000 Paired End 100 bases.

RNA reads acquired out of the murine xenograft models were quantified and categorized as mammalian or murine in the following manner: sequence counting then classification. The software Xenome was used to qualify gene grouping. The tool places the xenograft RNA sequences into two bins based on two reference genomes, HG19 (human) and MM10 (mouse). Consequently, reads classified as human were subsequently aligned and quantified. Alignment was conducted using the STAR aligner to the HG19 reference genome. Using featureCounts, a function within the Rsubread package, gene expression estimates were generated. Control samples were generated from the murine fat pad which had circa 90% mouse reads, whereas other models had about 90% human reads. RNA expression within PDX lines was represented as studentized fragments per kilobase of transcript per million mapped read values. Individual gene scores were generated using the thirty-six available PDX lines. The heat map's PDX order was fixed after an unstructured sorting of genes within the EMT network (*TWIST1*, *SNAI1* and *SIX1*). Species classification was conducted under Xenome 1.0.1. The data were computed with the help of Rsubread version 1.15.9 and plotted using the gplots package in R version 3.0.1.

**Statistics.** Prism software (v5.0; GraphPad) was used for most statistical analyses. Two-tailed unpaired Student's *T*-test was used when a pair of conditions was compared. One-way analysis of variance (ANOVA) non-parametric followed by Tukey's post test was utilized when multiple conditions were being compared and analysed. Two-way ANOVA followed by Bonferroni post test was used to compare drug treatment conditions over time and repeated measure Two-way ANOVA was used in the proliferation assays. Survival curves were estimated by the Kaplan–Meier method and compared using the log-rank test or the Wilcoxon test, whichever is appropriate. Specific analyses used for each experiment is described in the figure legends.

**Data availability.** The RNA-seq data has been deposited and the accession number is GSE97726. All other remaining data are available within the Article and Supplementary Files, or available from the authors upon request

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

## Acknowledgements

We thank M. Vincent for technical assistance and critical reading of this manuscript, and D. Drasin and other past and present Ford lab members for helpful suggestions and comments regarding this work. We thank the University of Colorado Comprehensive Cancer Center (P30CA046934), which supports core facilities including the Animal Imaging and Flow Cytometry shared resources used in this study. The PDX model study was supported, in part, by NIH/NCI grants R01CA127857, P50 SPORE CA50183 and U54CA149196 (to M.T.L.). The Dan L. Duncan Comprehensive Cancer Center (BCM, P30CA125123) funded the the Genomic & RNA Profiling Shared Resource. All other studies in this manuscript were funded by NCI grant R01CA095277 to H.L.F.

## Author contributions

D.N. conceived the ideas within the manuscript and performed the majority of the experiments within the manuscript, in addition to writing and editing the manuscript. H.Z. performed many of the co-culture and inhibitor experiments. M.U.J.O. performed the majority of analysis of human datasets examining clinical outcomes. X.Z., M.-F.W., S.G.H., L.M.S. and L.D.W. performed the PDX model study and RNA seq from the PDX models. D.M.H. and C.A.S. analysed the RNA-seq data and generated the heat map for PDX models. M.T.L. provided the Gli1-GFP reporter system, conceptualized and funded the PDX model study, provided crucial advice for some experiments and edited the manuscript. H.L.F. conceptualized the ideas and experimental questions and interpreted the data along with D.N., as well as edited the manuscript and funded the study.

## Additional information

**Competing interests:** M.T.L. is a manager in StemMed Holdings LP and a limited partner in StemMed Ltd. The remaining authors declare no competing financial interests.

**Publisher's note**: 

