## [Peer Review File · Nature Communications]

Reviewers' comments:

Reviewer #1 (Remarks to the Author):

In this manuscript, the authors address a relevant question in the area of tumor heterogeneity and metastasis, i.e. whether breast cancer cells, that have undergone an Epithelial-Mesenchymal Transition (EMT) via stable expression of EMT-transcription factors influence the properties of epithelial tumor cells. Thus, the premises for their studies are 1) within (breast) carcinomas, only a fraction of tumor cells undergo an EMT, leading to inter-tumor heterogeneity and 2) although the role of cells that have undergone an EMT as founders of macroscopic metastasis is highly contested, they may influence metastatic ability of epithelial carcinoma cells.

To this end, the authors conduct co-culture studies using in vitro transformed human mammary epithelial cells (HMLER), as well as the breast cancer cell line MCF7 stably transduced with EMT-transcription factors Twist1, Snail1 and Six1, respectively. They observe that Six1, downstream of Twist1 and Snail1, enhances migration and invasion of parental HMLER cells in a Gli1-dependent, non-cell autonomous manner through soluble factor(s), but independently of Hedgehog (Hh) ligands and Smoothed-activation. In MCF7 cells, they observe a Hh-dependent non-cell autonomous downregulation of E-cadherin and increase in anoikis resistance mediated by MCF7 cells overexpressing Six1. Supported by xenograft experiments the authors conclude that direct inhibition of Gli1, achieved by the small-molecule inhibitor GANT61, constitutes a strategy to generally block non-cell autonomous activation of Gli1 and thereby, impact tumor growth and, possibly, metastasis.

These findings are novel and of broad interest for research related to EMT, presented in a well-structured and –written manuscript that largely supports conclusions drawn. Given the fact that inhibitors of Hh-signaling targeting Smoothed have not shown promise for treating breast cancer, the observation that activation of downstream Gli1-signaling can occur in a Hh-ligand- and Smoothed-independent manner may be important for future therapeutic strategies.

Nonetheless, there are some serious experimental and conceptual issues, particularly with respect to data drawn from the MCF7 cells, that need to be addressed:

Figure 2k. The E-cadherin immunofluorescence stainings are of poor quality and not very convincing. The same is true for Figure 5f. and 5i. Here, it would be good to provide better quality images along with E-cadherin protein levels, as determined by Western blotting.

Figure 2j. – l. and all other MCF7-Six1 co-culture experiments: The authors show that MCF7-Six1 cells induce a partial EMT and increase anoikis resistance through an increase in SHH secretion, which correlates with 7Gli1-reporter activity. Is this effect really only paracrine, as the authors suggest? One would think that SHH secretion also results in autocrine stimulation of the MCF7-Six1 cells.

Figure 3. The authors show that overexpression of Eya2 induces SHH expression in HMLER-Six1 cells – do MCF7-Six1 cells, which secrete SHH express Eya2 endogenously?

Figure 4. How does 7Gli1-reporter activity compare between HMLER and HMLER-Snail1 and – Twist1 cells, i.e. is Gli1 also active in these cells, or is activation of Gli1-transcription purely a paracrine effect (see also comments for Figure 2j - l.)?

Figure 6. It is puzzling that the authors use MCF7/MCF7-Six1 cells for an experiment aimed to investigate metastasis, since all in vitro data showing a non-cell autonomous effect on migration and invasion were conducted with HMLER cells, which are likewise tumorigenic. Unless the authors show that MCF7-Six1 cells also increase migration/invasiveness in MCF7 cells in a non-cell autonomous manner, this choice of experiment is hard to follow. Also, it is not entirely clear

whether the luminescent/fluorescent signals shown in Figure 6b. and S6a. allow the discrimination between local tumor and metastasis, since much of the signal is merged with the primary tumor – it would have been much more convincing to measure luminescence/fluorescence in isolated organs and/or supplement these analyses by immunohistochemistry. Moreover, it should be more clearly stated that primary tumor growth is significantly affected by treatment with GANT61. Taken together, these data do not allow sound discrimination between an effect on primary tumor growth or metastasis.

Finally, in the text, MCF7-Six1 cells are referred to as EMT-cells. Why? Earlier on the authors stated that overexpression of Six1 does not induce an EMT in these cells.

The title also echoes this inaccuracy and should be revised. Finally, at least in the Discussion it would be useful to speculate how exactly Hh-ligand and Smoothed-independent activation of Gli1-transcription may be induced via soluble factors, since this is a question that remains unsolved.

Reviewer #2 (Remarks to the Author):

In this paper the authors use breast cancer cell lines to examine the ability of tumor line cells expressing EMT drivers to influence the migration, invasiveness, and gene expression patterns of neighboring non-EMT tumor line cells. They find that these non-autonomous effects occur and are mediated by the activation of the GLI transcription factors. In MCF-7 cells this appears to be mediated by canonical Hedgehog signaling whereas in HMLER cells a non-canonical pathway is used. These findings correlate with results from human tumors where high Six1 and Gli1 are associated with poor prognosis, and patient derived xenographs where high levels of GLI transcription factors better correlate with EMT transcription factor levels than Hedgehog ligands.

The data are generally of good quality and the conclusions supported.

All of the experiments look at EMT MCF-7 cells signaling to non-EMT MCF-7 or EMT HMLER cells signaling to non-HMLER cells. Can EMT MCF-7 cells signal to non-EMT HMLER cells and vice versa? Is the difference in signaling a consequence of just differences in the signaling cells or also the receiving cells?

Additional issues that should be addressed:

What is the nature of the signal that leads to non-canonical activation of GLI? The TGF-beta, KRAS and the mTOR/S6K1 pathways have all been shown to be able to activate GLI in a non-canonical fashion. In addition FOXC1 has been implicated in non-canonical activation of GLI in breast tumors. Are any of these pathways responsible for the results observed here?

A general concern is the use of tumor cell lines and how well their behavior correlates with tumors in vivo.

Minor issues:

The text description of 5h is confusing.

In figure 5i the first and third panels have identical labels.

We would like to thank the reviewers for their overall positive reviews and thoughtful comments and helpful suggestions. We believe that we have satisfactorily addressed the comments of the reviewers, and that addressing these points has indeed significantly improved our manuscript. We are thus submitting our revised manuscript, and have provided a point-by-point response to all the reviewer's comments below.

Reviewers' comments:

Reviewer #1:

- 1. Figure 2k (in revised manuscript, this is *Figure 2g*). The E-cadherin immunofluorescence stainings are of poor quality and not very convincing. The same is true for Figure 5f. and 5i (in revised manuscript, these figures are *5f and 5h*). Here, it would be good to provide better quality images along with E-cadherin protein levels, as determined by Western blotting.**

We have provided better quality images of E-Cadherin staining for the above-mentioned figures in the revised manuscript (figs. 2g, 5f, and 5h) as requested and have quantified membranous E-cad levels across multiple experiments with the conditioned medium. Because we are demonstrating differences in membranous E-cadherin, we felt that ICC was a much better way to demonstrate this effect, as opposed to western blot analysis.

- 2. Figure 2j. – I (Figures *2f-h in the revised manuscript*) and all other MCF7-Six1 co-culture experiments: The authors show that MCF7-Six1 cells induce a partial EMT and increase anoikis resistance through an increase in SHH secretion, which correlates with 7Gli1-reporter activity. Is this effect really only paracrine, as the authors suggest? One would think that SHH secretion also results in autocrine stimulation of the MCF7-Six1 cells.**

Indeed, the point made by the reviewer is a good one. We do see an autocrine effect on the MCF7-Six1 cells themselves. As we show in Fig. 2f-h (please note changed figure order in revised manuscript to adhere to the journal requirements) MCF7-Six1 cells have increased EMT (published previously¹), as well as increased anoikis resistance compared to MCF7-Ctrl cells. We also see that MCF7-Six1 cells have increased 7-Gli1 reporter activity compared to MCF7-Ctrl cells (shown below). If the reviewer thinks that the Gli1 data shown below is helpful to the manuscript, we would be glad to add it.

3. Figure 3. The authors show that overexpression of Eya2 induces SHH expression in HMLER-Six1 cells – do MCF7-Six1 cells, which secrete SHH express Eya2 endogenously?

Yes, MCF7 cells do express endogenous Eya2 (which has been previously published²) and these cells express higher levels of endogenous Eya2 than HMLER cells (see below). We have added this figure to the supplementary figures (Supplementary Fig. 3g) in the revised manuscript.

4. Figure 4. How does 7Gli1-reporter activity compare between HMLER and HMLER-Snail1 and – Twist1 cells, i.e. is Gli1 also active in these cells, or is activation of Gli1-transcription purely a paracrine effect (see also comments for Figure 2j - l.)?

We performed the Gli1 reporter assay experiment in the HMLER system (see below), and didn't observe increased 7Gli1 activity in either the HMLER-Snail1 or Twist1 cells when compared to the HMLER-Ctrl cells, suggesting that Gli1 activation is likely primarily through a paracrine effect on the HMLER-Ctrl cells when CM media is used from HMLER-Twist1 or Snail1 cells. We have not added this information to the manuscript, but can do so if the reviewer thinks it is helpful.

5. **Figure 6.** It is puzzling that the authors use MCF7/MCF7-Six1 cells for an experiment aimed to investigate metastasis, since all *in vitro* data showing a non-cell autonomous effect on migration and invasion were conducted with HMLER cells, which are likewise tumorigenic. Unless the authors show that MCF7-Six1 cells also increase migration/invasiveness in MCF7 cells in a non-cell autonomous manner, this choice of experiment is hard to follow.

We apologize that the above stated issues were not clear. In our manuscript, we demonstrate that Six1 is the key mediator of non-cell autonomous effects induced by EMT TFs, as it is critical both downstream of Twist1 and Snail1, as well as can mediate non-cell autonomous effects on its own. We have also previously published that introduction of Six1 into MCF7 cells causes BOTH an EMT and increases metastasis in a cell autonomous manner (in vivo in mouse models)¹. Hence, this system was utilized to demonstrate the non-cell autonomous functions of Six1 in an isolated manner (so that we did not also have to deal with cells expressing Twist1 and Six1 together or Snail1 and Six1 together).

*As mentioned in the text, while Six1 induces an EMT *in vitro* (as demonstrated by an alteration in molecular markers¹) and increases metastasis of MCF7 cells *in vivo*, it **does not** increase migration or invasion *in vitro*. These data suggest to us that the tumor microenvironment (which is absent in our *in vitro* experiments) may play a crucial role in the Six1-induced metastasis of the MCF7 cells (and thus necessitates *in vivo* experiments, such as those performed within our manuscript).*

6. **Also, it is not entirely clear whether the luminescent/fluorescent signals shown in Figure 6b and S6a allow the discrimination between local tumor and metastasis, since much of the signal is merged with the primary tumor – it would have been much more convincing to measure luminescence/fluorescence in isolated organs and/or supplement these analyses by immunohistochemistry.**

*In our *in vivo* luminescence and fluorescence measurements, while we did not perform ex-vivo imaging of the organs, we always compared the signal from tumors of equal volume and size, as measured using calipers and luminescence/fluorescence signals, and thus one cannot attribute the altered signal to primary tumor size (the differences must arise in that case from secondary sites). Our method of quantitation (IVIS measurement of luciferase or fluorescence signal) would be expected to be more accurate than immunohistochemistry because we did not observe an “all-or-nothing” signal, but rather an increase/decrease in “metastatic” signal from the tumor cells depending on the condition/treatment.*

7. Moreover, it should be more clearly stated that primary tumor growth is significantly affected by treatment with GANT61. Taken together, these data do not allow sound discrimination between an effect on primary tumor growth or metastasis.

GANT61 treatment was initiated on the tumors when they reached a volume of 1cm^3 . Contrary to what has been published in the literature, we did **not** observe a significant effect of GANT61 treatment on the **overall** primary tumor size or volume (see below- in response to this question we have added this figure to the revised manuscript as Supplementary Fig. 6g). The fact that we did not see a significant effect of GANT61 on primary tumor size could be due to: 1) The timing of the treatment initiation; 1cm^3 versus 0.2cm^3 (used in the literature) or because the tumor may have been comprised of a majority of MCF7-Six1-tRFP cells (which we show are not inhibited by GANT61), such that the inhibition of MCF7-Ctrl-luc cells in the primary tumor did not make a significant difference in the overall tumor burden. We did observe a significant effect on the luminescence signal from the MCF7-Ctrl cells specifically in both the primary tumor and in the metastases (what is being shown in Fig. 6 is the growth of only the MCF7-Ctrl cells within the primary tumor, not the overall growth or size of the tumor, as the overall growth would be a measurement of both Ctrl and Six1 cells). We have made this issue more clear within the manuscript and apologize for the confusion.

8. Finally, in the text, MCF7-Six1 cells are referred to as EMT-cells. Why? Earlier on the authors stated that overexpression of Six1 does not induce an EMT in these cells. The title also echoes this inaccuracy and should be revised.

As referred to above, we have previously published that MCF7 cells that overexpress Six1 (MCF7-Six1 cells) cell autonomously undergo EMT, as seen by an increase in Fibronectin and decrease in Cytokeratin-18, as well as re-localization of E-cad from the membrane to cytoplasm, compared to the corresponding MCF7-Ctrl cells that don't express Six1¹. Hence they are referred to as "EMT" cells in the manuscript, while the MCF7-Ctrl cells are referred to as "non-EMT" cells.

9. Finally, at least in the Discussion it would be useful to speculate how exactly Hh-ligand and Smoothened-independent activation of Gli1-transcription may be induced via soluble factors, since this is a question that remains unsolved.

As per this helpful suggestion we have included the following in the discussion section of the revised manuscript: “Several non-canonical mechanisms of GLI1/2 activation have been described in the literature including, but not limited to, mechanisms that promote transcription of GLI1/2, increase their stability, or regulate their cellular compartmentalization⁴. In addition, various secreted molecules have been linked to GLI activation⁴ in a non-canonical manner, which can be explored as potential means by which the EMT-TFs non-canonically activate Hh signaling. To date, we have not identified a known non-canonical pathway by which EMT TFs non-cell autonomously activate Hh signaling, and thus are also using unbiased approaches to answer this question”.

We have begun to elucidate novel non-canonical mechanisms of GLI activation (see response to reviewer #2, question #2 below), but we feel at this point it is too early to add this information into the current manuscript (as you will see from our response to below, the context and particular combination of EMT TFs may allow for different mechanisms of non-canonical GLI activation).

Reviewer #2 (Remarks to the Author):

In this paper the authors use breast cancer cell lines to examine the ability of tumor line cells expressing EMT drivers to influence the migration, invasiveness, and gene expression patterns of neighboring non-EMT tumor line cells. They find that these non-autonomous effects occur and are mediated by the activation of the GLI transcription factors. In MCF7 cells this appears to be mediated by canonical Hedgehog signaling whereas in HMLER cells a non-canonical pathway is used. These findings correlate with results from human tumors where high Six1 and Gli1 are associated with poor prognosis, and patient derived xenografts where high levels of GLI transcription factors better correlate with EMT transcription factor levels than Hedgehog ligands.

The data are generally of good quality and the conclusions supported.

- 1. All of the experiments look at EMT MCF-7 cells signaling to non-EMT MCF-7 or EMT HMLER cells signaling to non-HMLER cells. Can EMT MCF-7 cells signal to non-EMT HMLER cells and vice versa? Is the difference in signaling a consequence of just differences in the signaling cells or also the receiving cells?**

The question posed above is excellent. We thus performed this experiment by taking CM from Six1 cells and placing it on the HMLER-Ctrl cells. Importantly, we observed an increase in migration of HMLER-Ctrl cells cultured in CM from MCF7-Six1 cells compared to when cultured in MCF7-Ctrl CM. We have added this data to the supplementary figures (Supplementary Fig. 2j) in the revised manuscript. Unfortunately, we were unable to do the reverse experiment as the MCF7 cells would not grow in HMLER media (whether conditioned or not).

These data suggest that there is a difference in signaling molecules in the CM.

Additional issues that should be addressed:

- 2. What is the nature of the signal that leads to non-canonical activation of GLI? The TGF-beta, KRAS and the mTOR/S6K1 pathways have all been shown to be able to activate GLI in a non-canonical fashion. In addition FOXC1 has been implicated in non-canonical activation of GLI in breast tumors. Are any of these pathways responsible for the results observed here?**

[REDACTED]

- 3. A general concern is the use of tumor cells lines and how well their behavior correlates with tumors in vivo.**

We agree that it would be optimal to perform these experiments in genetically engineered mouse models in which EMT spontaneously occurs. However, such experiments would require us to devise models in which we could eliminate the EMT cells that spontaneously occur as they occur in vivo, and thus will be challenging to develop during the time frame required for resubmission of this manuscript (and will likely take years). In order to begin to approach the mechanistic aspects of crosstalk between EMT and non-EMT cells, we thus felt that the tumor cell line system was an ideal means to address the question. We did use PDX models to demonstrate our overarching findings that targeting canonical Hh signaling may not be fruitful, and that instead, targeting more downstream activators of the pathway (Gli1) is a more promising approach. We also used RNA-seq of human and PDX models to demonstrate a strong correlation of EMT-TFs and GLI1 (itself activated by the Hh pathway), and these data strongly suggest that our findings will be relevant to human tumors.

Minor issues:

The text description of 5h (*currently figure 5g*) is confusing.

In figure 5i (*currently figure 5h*) the first and third panels have identical labels.

We agree that the description is confusing and have rectified this issue in the manuscript (see page 9). We thank the reviewer for their astute observation of the mislabeling of the first and third panels of Fig. 5i (now figure 5h), and have corrected this labeling in the revised manuscript figures.

References

1. Micalizzi, D.S. *et al.* The Six1 homeoprotein induces human mammary carcinoma cells to undergo epithelial-mesenchymal transition and metastasis in mice through increasing TGF-beta signaling. *The Journal of clinical investigation* **119**, 2678-2690 (2009).
 2. Farabaugh, S.M., Micalizzi, D.S., Jedlicka, P., Zhao, R. & Ford, H.L. Eya2 is required to mediate the pro-metastatic functions of Six1 via the induction of TGF-beta signaling, epithelial-mesenchymal transition, and cancer stem cell properties. *Oncogene* **31**, 552-562 (2012).
 3. Wang, C.A. *et al.* SIX1 induces lymphangiogenesis and metastasis via upregulation of VEGF-C in mouse models of breast cancer. *The Journal of clinical investigation* **122**, 1895-1906 (2012).
 4. Gu, D. & Xie, J. Non-Canonical Hh Signaling in Cancer-Current Understanding and Future Directions. *Cancers* **7**, 1684-1698 (2015).
-

REVIEWERS' COMMENTS:

Reviewer #1 (Remarks to the Author):

The authors have carefully responded to all of the points raised in the review. The manuscript should be published as provided in the revised form.

Reviewer #2 (Remarks to the Author):

I have read the revised manuscript and the authors' response, and they have appropriately addressed the concerns that I had about the manuscript.